# Application of Graphene-Based Materials for Detection of Nitrate and Nitrite in Water—A Review

**DOI:** 10.3390/s20010054

**Published:** 2019-12-20

**Authors:** Daoliang Li, Tan Wang, Zhen Li, Xianbao Xu, Cong Wang, Yanqing Duan

**Affiliations:** 1College of Information and Electrical Engineering, China Agricultural University, Beijing 100083, China; 2China-EU Center for Information and Communication Technologies in Agriculture, China Agricultural University, Beijing 100083, China; 3Key Laboratory of Agricultural Information Acquisition Technology, Ministry of Agriculture, China Agricultural University, Beijing 100083, China; 4Beijing Engineering and Technology Research Center for Internet of Things in Agriculture, China Agricultural University, Beijing 100083, China; 5Business school, University of Bedfordshire, Luton LU1 3BE, UK; Yanqing.Duan@beds.ac.uk

**Keywords:** graphene, nitrite, nitrate, electrochemical sensing, graphene oxide, reduced graphene oxide

## Abstract

Nitrite and nitrate are widely found in various water environments but the potential toxicity of nitrite and nitrate poses a great threat to human health. Recently, many methods have been developed to detect nitrate and nitrite in water. One of them is to use graphene-based materials. Graphene is a two-dimensional carbon nano-material with sp^2^ hybrid orbital, which has a large surface area and excellent conductivity and electron transfer ability. It is widely used for modifying electrodes for electrochemical sensors. Graphene based electrochemical sensors have the advantages of being low cost, effective and efficient for nitrite and nitrate detection. This paper reviews the application of graphene-based nanomaterials for electrochemical detection of nitrate and nitrite in water. The properties and advantages of the electrodes were modified by graphene, graphene oxide and reduced graphene oxide nanocomposite in the development of nitrite sensors are discussed in detail. Based on the review, the paper summarizes the working conditions and performance of different sensors, including working potential, pH, detection range, detection limit, sensitivity, reproducibility, repeatability and long-term stability. Furthermore, the challenges and suggestions for future research on the application of graphene-based nanocomposite electrochemical sensors for nitrite detection are also highlighted.

## 1. Introduction

Nitrogen is one of the main factors causing eutrophication in water [1,2]; it can be divided into organic nitrogen and inorganic nitrogen. Among them, nitrate nitrogen and nitrite nitrogen are the forms of inorganic nitrogen, which are widely found in water [3]. They are mainly caused by the excessive use of agricultural fertilizers and the discharge of wastewater from human living and other production activities [4]. However, the presence of excessive nitrate and nitrite in water has a great impact on aquatic organisms and human health. 

When the content of nitrite nitrogen exceeds 0.02 mg/L, which is directly toxic to aquatic organisms, it affects their growth and development, induces outbreaks of diseases and even causes a large number of deaths. Nitrite in drinking water can irreversibly convert hemoglobin into methemoglobin, which results in a decrease in oxygen carrying capacity in the blood [5]. It can react with secondary amines to produce n-nitrosamines, which are likely to lead to gastric cancer [6]. Since nitrate is converted into nitrite in the digestive tract, it also very harmful to the human body [7]. It can also lead to the unnatural reproduction of aquatic plants and algae, leading to “red tides” and the death of fish [8,9]. The United States (U.S.) Environmental Protection Agency has limited the nitrite and nitrate content in drinking water to 1 mg/L and 10 mg/L [10] and other countries have made similar regulations. The concentration of nitrate and nitrite in water has become one of the indexes for evaluating water quality. Therefore, the detection of nitrate and nitrite in water samples at low concentrations has been the focus of research [11]. 

So far, many methods have been used to detect nitrate and nitrite, such as spectrophotometry [12,13,14], chemiluminescence [15,16], chromatography [17], fluorescence [18,19] and electrochemical methods [20,21,22]. Among them, the electrochemical method is one of the most commonly used detection methods because of its advantages of being low cost, fast, direct and high efficiency [23]. However, the sensitivity and accuracy of the electrodes are reduced because of disturbance by other substances. Bare electrodes need higher applied potential, which limits the application of bare electrodes [24]. Therefore, electrochemical techniques based on modified electrodes have been extensively studied due to their simplicity, high sensitivity and selectivity [25,26,27]. Using a suitable catalyst modified electrode can not only increase the catalytic current for nitrite and nitrate but also expand the detection range and reduce the detection limit.

At present, many nanomaterials have been developed to modify electrodes to improve the detection performance of nitrate and nitrite, such as metals [28,29,30,31,32], metal oxide [33], conductive polymers [34,35] and carbon nanomaterials [36,37,38], among which carbon nanomaterials are the most widely studied nanomaterials. Graphene is a two-dimensional carbon nanomaterial consisting of carbon atoms with sp^2^ hybrid orbital and it has a honeycomb hexagonal lattice network and extends infinitely [39]. It is widely used to modify electrodes of electrochemical sensors because of its unique features such as larger specific surface area, stronger mechanical strength, higher conductivity and higher electrochemical activity [40,41]. According to the different molecular structure of graphene, it can be divided into graphene, graphene oxide and reduced graphene oxide (as shown in Figure 1), different shape and structure with different performance. Graphene alone has limited performance, so it often synthesizes composite nanomaterials with metals, metal oxides, polymers and so forth, to enhance its electro catalytic ability [42].

This paper reviews the application of graphene-based materials in electrochemical detection of nitrate and nitrite in water. It analyzes the principle and performance of detecting nitrate and nitrite in graphene-based composites and summarizes the working conditions and performance of each sensor in the table, including working potential, pH, detection range, detection limit, sensitivity, reproducibility, repeatability and long-term stability. Finally, the advantages and disadvantages of various sensors and the future development trends are analyzed. The aim is to provide a systematic and comprehensive review of graphene-based nanomaterials electrochemical sensors for detecting nitrate and nitrite in water and to provide useful reference for future research.

## 2. Graphene-Based Nanocomposites

Graphene has no functional groups, which makes the molecular structure of graphene very stable. Due to its excellent electrical and mechanical properties, it is widely used in biomedicine, materials, energy and other fields. In sensor application, graphene material alone has limited use but it is often combined with other materials to improve conductivity and electrocatalysis. This section introduces the determination of nitrate and nitrite in water by graphene matrix composites. It then summarizes the performance of these nanocomposites in Table 1.

### 2.1. Metals

Metal nanoparticles are widely used in electrochemical catalysis because of their large specific surface area and good electro catalytic activity. They are divided into non-precious metals and precious metals; non-precious metals are mainly Fe, Cu, Co, and so on, precious metals are mainly Au, Ag and Pt and so on. The combination of metal and graphene has better electro catalytic activity.

Fe nanoparticles (FeNPs) are one of the excellent electro catalysts for nitrite oxidation. Mani et al. prepared graphene (Gr)/Multiwalled carbon nanotube (MWCNTs)/FeNPs (Gr/MWCNTs/FeNPs) nanocomposites modified glassy carbon electrode (GCE) for nitrite detection [43]. MWCNTs were added into graphene oxide (GO) dispersion solution; GO-MWCNTs were obtained by ultrasonic treatment for 2 h and then added FeCl_3_ and NaBH_4_ to get Gr-MWCNTs/FeNPs nanocomposites by stirring for 3 h. Under the condition of applied potential of 0.77 V and pH of 5, the linear range of nitrite determination with this modified electrode is 0.1 µM–1.68 mM and the minimum detection limit is 75.6 nM. The electrode has the characteristics of stability, repeatability and anti-interference. The recovery of detection in different water bodies is between 96.1% and 103.7%, which proves its practicability.

Compared with FeNPs, the sensitivity of Cu/graphene composite modified glassy carbon electrode for nitrite detection was improved. Majidi et al. prepared nano-porous copper films (NPCF) and graphene nanosheets (GNs) modified GCE for sensitive detection of nitrite [44]. NPCFs were prepared with a one-step chemical method and then NPCF was assembled onto the GNs/GCE surface by the potentiostatic method. Due to the synergistic effect of NPCF and GNS, the electro catalytic activity of NPCF-GNs/GCE electrodes for nitrite oxidation was significantly improved. Under the condition of applied potential of 0.8 V and pH of 9, the detection range, sensitivity and detection limit of the modified electrode are 0.1–100 µM, 3.1 µAµM^−1^cm^−2^ and 0.088 µM, respectively. The response speed of the electrode is faster (3 s) and the selectivity of the electrode is better than that of a bare electrode. The detection results in actual river and lake water are close to those of spectrophotometry.

Metal Cu also shows good electro catalytic activity for nitrate reduction. Wang et al. prepared self-assembled graphene decorated with Cu nanoparticles for nitrate detection [45]. Self-assembled graphene overcomes the disadvantages of poor adhesion of a traditional connection between graphene and electrodes and provides a three-dimensional network structure for Cu nanoparticles. This composite has high sensitivity because Cu has a catalytic effect on nitrate reduction. The nanocomposite used in a three-electrode system sensor detected a nitrate concentration range of 10–90 µM and a detection limit of 7.89 µM. The detection result of the method in the actual lake water is consistent with the results of the standard method. The developed sensor has a low cost, small size and high sensitivity.

The application of cobalt nanoparticles in modified electrodes can improve the electro catalytic activity. Wang et al. prepared cobalt nanoparticles (CoNPs), poly (3, 4-ethylenedioxythiophene) (PEDOT) and graphene (Gr) nanocomposites modified GCE for nitrite detection [46]. Graphene solution dried up and fixed on the surface of GCE to form Gr-GCE. PEDOT was fixed on the surface of Gr-CCE by the same method. Cobalt nanoparticles were electrodeposited onto PEDOT-Gr/GCE by cyclic voltammetry. CoNPs-PEDOT-Gr/GCE has high electro catalytic activity for nitrite oxidation, mainly because graphene provides high conductivity and CoNPs, highly dispersed on the graphene surface through PEDOT, can provide fast mass transfer. Under the optimum conditions of pH 6.5 and working voltage 0.45 V, the linear concentration range of nitrite detected by this electrode is 0.5–240 µM and the detection limit is 0.15 µM (S/N = 3), which has high stability and repeatability. However, 20-fold ascorbic acid, 10-fold S^2−^, S_2_O_3_^2−^ and I^−^ may serious affect the detection results of this electrode.

These studies have been carried out under different conditions and conditions, Fe, Cu, Co combined with graphene showed excellent electro catalytic ability. Hameed et al. studied the effects of Cu, cobalt and nickel on the electro catalytic activity of nitrite under the same structure and the same binding conditions. They synthesized three core-shell structure nanocomposites of Cu@Pt/Gr (show Figure 2) [47], Ni@Pt/Gr [48] and Co@Pt/Gr [49] using different methods. Among them, Cu, Co and Ni are all core layers, Pt is shell layer and graphene is bottom conductive material. Under the same conditions (Applied potential is 0.85V, pH is 4), their detection ranges are 1 µM–15 mM, 0.01–15 mM, 1 µM–15 mM, detection limits are 0.035 µM, 0.49 µM and 0.145 µM, respectively. Among them, Cu@Pt/Gr has the lowest detection limit for nitrite, Co@Pt/Gr has the highest sensitivity for nitrite and Ni@Pt/Gr has the largest detection range. They all have good selectivity, stability and reproducibility and the detection recovery rate in actual tap water and river water is satisfactory.

Gold nanoparticles (AuNPs) exhibit excellent catalytic properties for nitrite oxidation. Li et al. synthesized AuNPs and graphene sulfide (SG) nanocomposites, which modified glassy carbon electrodes to detect nitrite concentration [50]. Firstly, graphene sulfide was deposited on the glassy carbon electrode and then AuNPs were adsorbed on the electrode by electrostatic attraction to obtain AuNPs/SG/GCE. At a voltage of 0.73 V, the electrode detect nitrite in the range of 10 µM to 3.96 mM and the sensitivity is 0.454 µAµM^−1^. The detection limit of this method is 0.2 µM and it has good reproducibility and stability. The measured values in actual river water are consistent with spectrophotometric method.

Wang et al. constructed a cheap, paper and disposable nitrite electrochemical detection platform based on AuNPs/Gr/mixed cellulose ester(MCE) [51]. AuNPs were assembled on Gr/MCE electrodes by electrodeposition. Due to the change of mass transport mechanism of AuNPs/Gr/MCE electrodes, the voltammetry response of AuNPs/Gr/MCE electrodes is much higher than that of traditional GCE and industrial electrodes. Under the condition of 0.74 V and the pH of 4.75, the linear concentration range of nitrite is 0.3–720 µM and the limit of detection is 0.1 µM. The detection of nitrite concentration in Lake and river water by this sensor is similar to that by ultraviolet-visible method. The advantage of disposable AuNPs/Gr/MCE electrode is that it does not need to consider the fouling caused by the oxidation products adsorbed on the surface.

Different structures have different electro catalytic effects. He et al. studied the electro catalytic effect of one-step hydrothermal preparation of wavy graphene(w-Gr) and Au nanocomposites [52]. Compared with bare glassy carbon electrode and Gr/GCE, w-Gr/Au/GCE has lower resistance, which indicates that it has strong electron transfer ability. Owing to the 3D structure of w-Gr and the synergistic effect of w-Gr and Au, w-Gr/Au has better electro catalytic ability. Under the condition of applied voltage of 0.84 V and pH of 7.2, the detection linear range of the electrode is 0.1–1000 µM and 1000–5000 µM and the detection limit is 0.06 µM. The practicability of the electrode has been verified in tap water.

Li et al. synthesized three-dimensional composites for nitrite detection using graphene [53], AuNPs, MnO_2_ and carbon nanotubes. In this 3D composite, Gr/CNTs form a three-dimensional high conductivity network for fast electron transport, mass diffusion and structural stability. AuNPs provide not only the conduction pathway of electron transfer but also the active site of nitrite catalysis. In the experimental environment with applied potential of 0.8 V and pH of 7, the linear range of detection is 1–2896 µM and the detection limit is 0.05 µM. The recovery rate of detection in drinking water and lake water is 98.6%–102%, which proved that the material can be used in practical samples.

Based on the review of the current, it is found that the electro catalytic effect was greatly improved because of the synergistic effect of metal and graphene materials. These modified electrodes have lower detection limit and higher sensitivity, which improves the performance of electrochemical sensors. However, the metal particle size will cause the change of surface volume ratio, which has a certain impact on the catalytic effect. In the future, it is necessary to study and determine the optimal metal particle size.

### 2.2. Polymer

Polymers include conductive polymers and polyelectrolytes. Conductive polymers/graphene composites can enhance conductivity and sensitivity. Polyelectrolytes/graphene can attract anions and enhance sensitivity. In addition, the polymer has low cost and is easy to form film, which is suitable for modifying electrode.

Polypyrrole (PPy) is a common conductive polymer with good conductivity and its hybridization with graphene can enhance conductivity. Ye et al. prepared graphene (Gr)/PPy/chitosan(CS) nanocomposite film modified on a glassy carbon electrode for nitrite [54]; PPy plays the role of electrochemical oxidation of nitrite and the surface of CS has abundant amino groups and is attractive to nitrite. The Figure 3 show the synthesized progress of Gr/PPy/CS nanocomposites. The modified electrode performs well in nitrite detection, the linear detection range is 0.5–722 µM, the detection limit is 0.1 µM and the reproducibility is 1.8%. Satisfactory results have also been obtained in actual lake and river water samples. 

Xiao et al. also prepared carboxyl graphene (CG)/PPy/CS nanomaterials for nitrite detection [55]. Their research optimized the amount of CG, the ratio of Py to CG, the volume of composite materials added and the pH value of the buffer. Under the optimum conditions, the nanomaterial modified GCE has a wide range and a low detection limit (0.2–1000 µM and LOD of 0.02 µM) for nitrite detection. Finally, the practicability of the electrode was verified in tap water.

Poly (3,4-ethylenedioxythiophene) (PEDOT) is a widely used conductive polymer. Nie et al. proposed a simple method to synthesize PEDOT/Gr nanocomposites and electrodeposit modified GCE for electrochemical detection of nitrite [56]. 3,4-ethylenedioxythiophene (EDOT)/graphene oxide (GO) was obtained by mixing EDOT with GO by ultrasound, then inserted into glassy carbon electrode and electrodeposited by cyclic voltammetry (CV). The oxidation current of nitrite was measured at the working potential of 0.78 V and the pH of 6. The linear range of detection is 0.3–600 µM and the detection limit is 0.1 µM. The electrode has good repeatability, stability and selectivity and the detection results in real river water are satisfactory. 

Tian et al. synthesized one-dimensional PEDOT-Gr nanocomposites on graphene nano sheets by in-situ electro polymerization of 3,4-ethylenedioxythiophene [57]. This 1D PEDOT structure has a large specific surface area and unique mixing structure, which makes PEDOT-Gr composite exhibit high electro catalytic activity. PEDOT-Gr composite coated with Ta film was used as working electrode to detect nitrite. Under the condition of applied potential of 0.7 V and pH of 7, the linear detection range was 20–2000 µM and the detection limit was 7 µM. In addition, the electrode has selectivity, fast reaction speed and long service life.

Liu et al. prepared poly (diallyl dimethyl ammonium chloride) (PDDA) coated Fe_1.833_(OH)_0.5_O_2.5_ and N-doped graphene(NG) nanocomposites(Fe_1.833_(OH)_0.5_O_2.5_/ PDDA/NG) by one-pot hydrothermal method, the Fe_1.833_(OH)_0.5_O_2.5_/PDDA/NG nanocomposites modified GCE for nitrite detection [58]. The results of Fe_1.833_(OH)_0.5_O_2.5_/PDDA/NG electrochemical performance test showed that the function of PDDA was to adsorb NO_2_^−^, NG serves as the conduction platform and Fe_1.833_(OH)_0.5_O_2.5_ was used as catalyst for NO_2_^−^ oxidation. When the applied potential is 0.837 V and the pH is 7.4, the linear range of the sensor for nitrite detection is 0.1–347 µM and 347–1275 µM and the detection limit is 0.027 µM. The average recovery of nitrite in tap water is 98.8%. 

Alahi et al. prepared a graphene sensor for nitrate detection [59]. The graphene monolayer was first recorded on the polyimide film and then the induced graphene electrode was transferred to Kapton tape to form a sensor patch. They also studied the effect of temperature on test results and set up temperature compensation calibration. The linear detection range of nitrate is 1–70 ppm in water samples. The results of water samples with unknown concentration are similar to those of laboratory methods and the error is less than 5%. The sensor has high sensitivity, fast response and good repeatability. The sensor also has Wi-Fi function, the data can be transmitted to the Internet of Things. It is suitable for real-time monitoring of nitrate concentration.

The polymer has good biocompatibility, stability and conductivity, which makes it easy to combine with graphene and improve the stability and conductivity. The electrocatalytic performance of polymer/graphene nanocomposites for nitrite and nitrate is improved under the synergistic effect of polymer and graphene and the catalytic performance of polymer/graphene nanocomposites for nitrite and nitrate is improved significantly.

### 2.3. Metal Compounds

Metal compounds include metal oxides, polyoxometalates, metal complexes and metal sulfides. These metal compounds have relatively low cost, relatively large surface area and good electro catalytic ability. Their hybridization with graphene shows good performance in the detection of nitrate and nitrite in water.

Li et al. prepared phosphotungstic acid (PW_12_O_40_^3−^)/chitosan(CS)/Gr nanocomposites by electrodeposition of PW_12_O_40_^3−^ on CS/Gr nanomaterials, which modified cysteamine(CS)/Au electrode(AuE) for nitrite detection [60]. They proved that the modified electrode has strong electrochemical and electro catalytic abilities through cyclic voltammetry and chronoamperometry. Under the condition of −0.18 V potential, the linear range of the modified electrode for nitrite reduction detection was 0.18 µM to 6.74 mM and the minimum detection limit was 0.02 µM. The electrode also has high selectivity, repeatability, reproducibility and stability.

Kung et al. synthesized metal–organic frameworks (MOF-525)/Gr nanocomposites by thermal growth of MOF-525 in dispersed Graphene nanoribbons (GNRs) suspension [61]. MOF-525 mainly catalyzes NO_2_^−^ and GNRs act as conductive bridges. The prepared MOF-525/GNRs nanocomposites were simply deposited on conductive glass substrates to form nitrite detection electrode. When the applied potential is 0.85 V, the linear range of the sensor for nitrite detection is 100–2500 µM, the detection limit is 0.75 µM and the sensitivity is 93.8 µAmM^−1^cm^−2^.

Wu et al. prepared copper sulfide (CuS) and nitrogen-doped graphene (NG) composites using the hydrothermal method [62]. The NG-CuS modified GCE for nitrite detection. Compared with bare GCE, NG/GCE and CuS/GCE, NG-CuS/GCE have a large specific surface area and high conductivity and have the best electro catalytic effect on nitrite oxidation. Under the optimum experimental conditions (the applied potential is 0.89V, the pH is 6), the response of NG-CuS/GCE to nitrite is stable within 3 s. The linear relationship of nitrite detection by modified electrode is 0.1 µM–14.02 mM, the detection limit is as low as 33 nM. The modified electrode also shows good stability, reproducibility and selectivity and has been successfully applied in lake water samples for nitrite detection.

Compared with noble metals and polymers, the cost of metal compounds is much lower. In addition, the combination of metal compounds and graphene also improves the electrocatalytic activity of graphene-based composites.

### 2.4. Others 

There are also other studies on graphene materials for nitrate and nitrite detection, such as ion-selective electrodes, graphene materials with different morphologies, graphene and other compounds and so forth.

Graphene modified copper electrode has good catalytic effect on nitrate reduction. Oznuluer et al. prepared a graphene modified copper electrode (Gr/CuE) by chemical vapor deposition (CVD) [63]. The electrochemical reduction of nitrate by graphene modified copper electrode in strong alkaline solution was studied by cyclic voltammetry and controlled potential method. In alkaline solution with pH of 12, it was found that the reduction peak density of composite electrodes in reducing nitrate was higher than that of bare copper electrodes, the linear range of nitrate detection is 9.0 × 10^−6^–9.4 × 10^−4^ and the detection limit is 10 µM. Moreover, the electrode has a fast response speed (less than 5s) and a sensitivity of 173.7 µAmM^−1^cm^−2^. This study shows that graphene modified electrode can be used in electrochemical studies but this study does not pay attention to stability studies [64]. 

The combination of graphene aerogels (GAs) and nitrate selective membrane has good selectivity for nitrate detection. Yan et al. uses GAs and nitrate ion selective membrane modified GCE for nitrate selective detection [65]. NO_3_^−^ ion selective membrane, graphene aerogel as the switching layer and glass carbon electrode as substrate. GAs have three-dimensional structure, which can effectively regulate the ion electron conversion between substrate and GCE of the ionophore. They used cyclic voltammetry to characterize the electrochemical behavior of the electrode and studied its performance in nitrate solution by potentiometric water layer test and potentiometric measurement. The experimental results show that the detection range of nitrate is 10^−4^ to 10^−1^ ML^−1^, the detection limit is 10^−4.2^ ML^−1^ and the electrode has good selectivity and long-term stability.

Palanisamy et al. added Gr into cyclodextrin (CD) solution and dispersed Gr/CD composites by simple ultrasound [66]. Graphene and cyclodextrin composites modified screen-printed carbon electrode(SPCE) were prepared for the detection of nitrite. Under the working voltage of 0.81 V and pH of 5, the modified electrode exhibited the highest electro catalytic activity for nitrite. The detection range of the electrode is 0.7 µM to 2.25 mM, the detection limit is 476.25 and the sensitivity is 476.25 µAmM^−1^cm^−2^. The electrode has selectivity, repeatability and reproducibility. The average recovery of nitrite in actual drinking water and tap water was 98% and 95.3%.

Some scholars have studied the electro catalytic effects of graphene with different graphene structures or doped with different elements, such as graphene sheets, graphene nanoribbons, electrochemical reduction of hollow graphene, three-dimensional holey graphene, N-doped graphene.

Ozturk et al. studied the graphene sheet modified GCE (Gr/GCE) for the detection of nitrite [67]. The graphene sheet was prepared by two-step electrochemical process in sodium dodecyl sulfate (SDS) solution. When the working potential was 0.805 V and the pH was 5, the Gr/GCE showed a linear range of 1–250 µM, a sensitivity of 0.843 µAmM^−1^cm^−2^ and a detection limit of 0.24 µM. The electrode has the advantages of high stability, fast response, high selectivity and reuse and the detection results in actual tap water are similar to those of spectrophotometry.

The research seems to show that graphene nanoribbons (GNs) are more sensitive to nitrite oxidation than graphene sheets [68]. Mehmeti et al. used GNs modified GCE for nitrite detection. Compared with the bare GCE, the peak oxidation current of GNs/GCE electrode increases and the peak potential decreases. Under the optimum conditions of 0.9V working potential and pH 3, the linear range of nitrite detection is 0.5–45 µM and 45–105 µM, the sensitivity is 6.3219 µAµM^−1^cm^−2^ and the detection limit is 0.22 µM. The stability, repeatability and selectivity of the electrodes were also verified. The electrode was used to analyze nitrite concentration in tap water samples and satisfactory recovery was obtained. However, the error of detection result of the electrode in the environment with high sulfite ion is large. 

Zhang et al. studied that the electrochemical detection of nitrite with electrochemical reduction of holey graphene(ERHG) has a larger linear range [69]. ERHG-based surface has a large number of nano-holes, which contain more exposed edge surfaces and high-density defects, it can significantly improve the electron and mass transfer of electrodes. The electro catalytic activity of ERHG modified GCE for nitrite oxidation was improved. The linear detection range was 0.2 µM to 10 mM with sensitivity of 0.311 µAµM^−1^ cm^−2^ and detection limit of 0.054 µM at applied potential of 0.92V and pH = 7.4. The sensor has excellent anti-interference, repeatability and reproducibility and its ability to detect nitrite in tap water and drinking water has been verified.

Compared with the graphene studied above, the 3D graphene has a lower detection limit. Chen et al. studied the preparation of three-dimensional holey graphene (3D-HG) by hydrothermal method for the sensitive detection of nitrite [70]. As 3D-HG has larger surface area and more exposed edge planes, its electrochemical active sites and electron transfer rate have been significantly improved. The GCE modified by 3D-HG has good electro catalytic performance for nitrite oxidation. When the applied potential is 0.819V and the pH is 7.4, the linear range of nitrite detection by differential pulse voltammetry is 0.05–500 µM and 0.5–10mM and the detection limit is 0.01 µM. The modified electrode also has good anti-interference, stability and reproducibility and performs well in actual tap water.

Li et al. used electrospinning, carbonization and hydrothermal methods to prepare n-doped graphene quantum dots (NGQDs) modified N-doped carbon nanofibers (NCNFs) composite for highly sensitive electrochemical detection of nitrite [71]. NGQDs have large surface volume ratio, many defects in surface and edge structure but their conductivity is poor. NCNFs produce a large number of defect sites, which is conducive to rapid electron transfer. The composite materials combined with NGQDs have many advantages, such as abundant defect sites and large electroactive surface area. Under the optimum conditions of applied potential 0.822V and pH 4.5, the linear range of the modified electrode for nitrite detection is 5–300 µM and 400–3000 µM and the detection limit is 3 µM. The electrode has good detection results in actual lake water and tap water. 

Graphene has stable structure and excellent conductivity but the electrode has limited performance only graphene modified. The combination of graphene with metals, polymers, metal compounds and other materials has significantly improved its electrical conductivity and electro catalytic performance in detecting nitrite and nitrate in water. In addition, the bonded nanocomposites have good selectivity in conventional samples. However, graphene itself has very good stability without functional groups, so it is difficult to bind with other substances. Besides, the performance of electrochemical sensors is not only limited by detection limit and detection range but also repeatability, stability and selectivity. At present, the repeatability and stability of these nanocomposites modified electrodes cannot ensure the real time on-line monitoring of the electrochemical sensors.

## 3. GO-Based Nanocomposites

Graphene oxide (GO) is an oxide of Gr, which destroys the bonding network of graphene’s sp^2^ and is a conductive insulator. The most widely used synthetic method of GO was proposed by Hummers [72]. The synthesis steps are shown in Figure 4. The surface of GO is rich in epoxy groups and hydroxyl groups and carbonyl and carboxyl groups on the edges, which improve the properties of GR layer and significantly improve the hydrophilicity of Gr layer. As GO is an insulator, its direct application in electrochemical sensing is limited. However, the oxygen-containing functional groups in GO are very beneficial to chemical modification or functionalization, so composite materials are often synthesized with other materials for electrochemical sensing.

### 3.1. Metals 

GO has no conductivity, so it is generally necessary to combine with conductive substances for electrochemical detection. Nano-metals have good conductivity and electro catalytic activity and their composite with GO solves the problem of non-conductivity of graphene oxide very well. Among them, due to the excellent electro catalytic ability and large surface volume ratio of silver nanoparticles (AgNPs), there are many studies on electro catalytic activity.

Platinum nanoparticles have good catalytic performance for nitrite oxidation. Bai et al. studied the morphology controlled synthesized of Pt nanoparticles on graphene oxide surface by gas-liquid interface reaction [73]. Aggregation-like, cube-like and globular platinum nanoparticles were synthesized by controlling the morphology of platinum nanoparticles through adjusting the reaction temperature under the protection of chitosan. The results of current-concentration studies show that the three Pt/GO modified GCE have good performance (as shown in Table 2). However, the applied potential of these three electrodes is relatively high.

AgNPs and GO composites are the most widely used in the detection of nitrate and nitrite in water. Ikhsan et al. prepared GO-Ag nanocomposites through deposited AgNPs on GO thin films by used garlic extracts and sunlight [74] and the nanocomposites modified GCEs (GCE) as electrochemical sensors for the detection of nitrite ions. At the voltage of 0.94V and a pH of 7.2, linear sweep voltammetry (LSV) and Amperometry i-t curves were used for the detection of nitrite by the GO-Ag/GCE. The linear range of the two methods is 10–180 µM and 1–1000 µM, with detection limits of 2.1 µM and 0.037 µM respectively. The electrodes are also selective and show good recovery in actual lake water samples. However, the stability and repeatability of the electrode have not been tested. 

Zhao et al. studied the controllable synthesis of AgNPs/GO composite nanomaterials by gamma irradiation [75]. The AgNPs here are spherical and the conductivity of GO-Ag is better than that of RGO. In addition, it was found that the best GO-Ag4 composites were obtained by adding 0.5 g polyvinyl-pyrrolidone (PVP) to the system without isopropanol (IPA) in the process of GO-Ag fabrication. The detection limit of nitrite is 0.24 µM and the detection range is from 1 µM to 1000 µM.

The combined of Ag and metal oxides with GO shows lower detection limits for nitrite. Li et al. prepared Ag-Fe_3_O_4_-GO nanocomposites by using Fe_3_O_4_-NH_2_ magnetic nanoparticles, GO and AgNO_3_ [76]. This material modified GCE was used to detect nitrite. At pH 6 and applied voltage 0.8 V, the electro catalytic effect of the electrode on nitrite is the best. The results show that there is a linear relationship between oxidation current and nitrite concentration in the range of 0.5–720 µM and 720–8150 µM with the sensitivity is 1.966 and 0.426 µAµM^−1^cm^−2^, respectively and the limit of detection is 0.17 µM. In addition, the electrode has stability, anti-interference and reproducibility. The recoveries detected in actual tap water ranged from 98.4% to 104.2%.

Hybridization of Ag and organic compounds with GO has also been used to detect nitrite. Ma et al. prepared ethanolamine (AE) and nano-Ag bifunctional GO (fG) composites (Ag-AEfG) by the one-pot hydrothermal method for the highly sensitive detection of nitrite [77]. AE is mainly used as an efficient reductant and dispersant to functionalize GO. The size and distribution of AgNPs on the surface of AEfG can be controlled by controlling the reaction temperature. Because of the nitrogen doping of Ag-AEfG and the synergistic effect between them, Ag-AEfG composites have good conductivity and electro catalytic activity. When applied voltage is 0.85V and pH is 7.4, the modified electrode has a good detection range of 0.05 µM–3 mM and limit of detection is 0.023 µM. The electrode can be detected in actual tap water.

Similarly, Ag has excellent electro catalytic ability for nitrate reduction. Shadfar et al. synthesized nanocomposites consisting of AgNPs, nanocellulose and GO(Ag/NC/GO-GCE) for sensing nitrate ion in environmental samples [78]. GO, NC and Ag catalysts have synergistic effects on the determination of nitrate. The effects of scanning rate, pH and nitrate concentration on the scanning results were studied and the optimum scanning conditions were obtained. The optimum linear detection range is 5 µM–10 mM and the detection limit is 0.016µM (S/N=3). The nanometer sensor has good sensitivity, selectivity and stability for the determination of nitrate in aqueous solution and has no obvious interference.

Other metals, such as titanium, have a certain catalytic effect on the reduction of nitrate. Ma et al. used treated GO dispersion and preprocessed Ti substrates to prepared Ti-GO electrodes with the hydrothermal method [64]. They did characterization experiments on Ti-GO electrode electrodes and electrochemical experiments on nitrate solution, the reduction efficiency of Ti, Cu, Cu-Zn and Ti-GO electrode for nitrate were also compared. Characterization experiments of the Ti-GO electrodes showed that the activity of the electrodes increased and the electrodes were stable in nitrate reduction and the Ti-GO electrode more stable and efficient in nitrate reduction experiments.

The composite of AuNPs and GO has also been studied. Rao et al. using pyrene methyl amine (PMA) as coupling agent to prepared GO-MWCNTs-PMA-AuNPs nanocomposites through a simple self-assembly method [79]. When the working potential is 0.73 V and the pH is 6.0, the nanomaterial modified GCE has the best electro catalytic activity for nitrite. Under the optimum conditions, the linear range of nitrite detection is 2 µM–10 mM, the detection limit is 0.67 µM and the sensitivity is 0.484 µAµM^−1^cm^−2^. The electrode has good reproducibility, stability and anti-interference. The recovery of nitrite in tap water is 99.1% to 101.2%.

Mo et al. modified GO-chitosan (CS) mixed film on GCE, then electrodeposited AuNPs on the surface of GO-CS/GCE by electrodeposition method [80]. The Au/GO-CS/GCE was used for the highly sensitive detection of nitrite. Because GO has a large surface area, CS is a polymer with positive charge. AuNPs enhance the contact area of catalytic reaction and provide a way for the electron transfer and catalytic process of GO-CS composites, GO-CS-AuNPs/GCE has good electrochemical catalytic activity for the detection of NO_2_^−^. When the potential is 0.8 V and the pH is 5, the linear detection range of the modified electrode is 0.9 to 18.9 µM and the detection limit is 0.3 µM.

Halloysite nanotubes have many advantages, such as natural nanotube structure, high aspect ratio (L/D), low cost, which are often used as substrates in modified electrode materials. The GCE modified by Au-HNTs-GO composite showed good catalytic activity for nitrite oxidation [81]. Under the condition of applied potential of 0.8V and pH of 6, the modified electrode has good performance in nitrite detection. In addition, the modified electrode has good reproducibility, long-term stability and anti-interference.

The combination of metal and GO improves the electrochemical activity, mainly because of the good conductivity provided by the metal and the catalytic activity of GO. Similarly, metal/GO nanocomposites with better electrochemical activity can be obtained if the size of metal nanoparticles is taken into account.

### 3.2. Others

Other GO-based materials have also been studied for the determination of nitrate and nitrite in water, such as GO/polymers, GO/metal oxides, GO/organic matter.

Polyaniline (PANI) is a conductive polymer with good electro catalytic activity and its combination with GO shows excellent electro catalysis for nitrite oxidation. Sivakumar et al. prepared PANI/GO nanocomposites by simple synthesis method for nitrite detection [82]. PANI has high conductivity and electrochemical redox performance, GO has high surface area and high electro catalytic activity. The PANI/GO modified GCE showed higher electro catalytic activity for nitrite oxidation than the PANI/GCE, GO/GCE and bare electrodes. Under the condition of applied potential is 0.83 V and pH is 5, the linear concentration range of nitrite detected by the modified electrode is from 2 µM to 44 mM, the sensitivity is 117.23 µAmM^−1^cm^−2^ and the detection limit is 0.5 µM. The modified electrode is suitable for actual rainwater and tap water.

Metal compounds have lower cost, better conductivity and electro catalytic ability and mixed with GO to modified electrode has better detection ability for nitrite. Muthumariappan et al. prepared Mn_3_O_4_ micro cubes (Mn_3_O_4_MC)/GO nanocomposites by the hydrothermal method and the Mn_3_O_4_MC/GO nanocomposites modified screen-printed electrode was used to detect nitrite [83]. The electrochemical activity of the modified electrode to nitrite was obviously improved and the oxidation peak current increased. The linear range of the modified electrode for nitrite detection is 0.1–420 µM (2.37 µAµM^−1^cm^−2^) and 420–1318 µM (1.23µAµM^−1^cm^−2^) and LOD is 0.02 µM. Since the unique properties of Mn_3_O_4_, the sensor has good sensitivity, selectivity and reproducibility. The sensor has been proved to be suitable for analysis in real water environment.

Jaiswal et al. succeeded in synthesizing dendritic MnO_2_ nanoparticles decorated GO nanocomposites by simple mechanical stirring and this material was used to make screen printing electrodes(SPE) for nitrite detection [84]. Under the condition of applied potential of 0.55 V and pH of 7.4, the linear range of detection is 0.1–1000 µM and the detection limit is 0.09 µM. The screen-printing electrode has good selectivity, stability and practicability.

GO combined with Fe_2_O_3_ or Prussian blue (PB) has a good catalytic effect on nitrite oxidation. Adekunle et al. first prepared Fe_2_O_3_ and Prussian blue nanoparticles and then modified GO and nanoparticles onto platinum electrode by drop-dry method [85]. The detection limits (sensitivity) of Pt-GO-PB and Pt-GO-Fe_2_O_3_ electrodes for nitrate are 16.58 µM (0.0093 µAµM^−1^) and 6.6 µM (0.0084 µAµM^−1^), respectively. These two electrodes also have selectivity and fast catalytic rate but the oxidation current of the electrodes decreases obviously after 20 times of operation.

Rostami et al. synthesized Fe_3_O_4_/GO/COOH nanocomposites for the detection of nitrite concentration [86]. The composite modified GCE has lower charge transfer resistance and better conductivity. By optimizing the working conditions (Ph = 4) and DPV method, the linear range of nitrite detection was 1–85 µM and 90–600 µM and the detection limit was 0.37 µM. The electrode has good selectivity and anti-interference and can be applied to practical drinking water.

Liu et al. prepared FEPA-GO by covalently grafted (4-ferrocenylethyne) phenyl amine(FEPA) on GO surface and then chitosan and FEPA-GO modified the GCE for sensitivity electrochemical determined nitrite [87]. Under the condition of applied potential of 0.756 V and pH of 6.8, the detection ability of the electrode for nitrite was tested. The results show that the linear detection range is 0.3–3100 µM and the detection limit is 0.1 µM. The electrode also has good reproducibility, stability and selectivity. The relative standard deviation of detection in tap water is 2.07%–3.15%. It shows that the electrode has practical value.

The catalytic efficiency of Gem-surfactant (Gem) -GO-polyoxometalate (POM) nanocomposites for nitrite oxidation is obviously improved [88]. Gem was used as stabilizer and linker to make POM nanoparticles uniformly distributed on graphite oxide sheets, which increased the active surface area of POM and enhanced the catalytic activity for nitrite oxidation with the catalytic efficiency of 54.6%. The linear detection range of the nanomaterial modified electrode is 5–500 µM and the detection limit is 0.39 µM.

The combination of metalloporphyrin and GO for the determination of nitrite in water with good result. Li et al. successful synthesized 5-(4-aminophenyl)-10, 15, 20-triphenylporphyrin Mn_(III)_(MnNH_2_TPP) and GO composite materials and modified GCE with drop casting method [89]. The detection ability of the electrode for nitrite in water was studied by cyclic voltammetry and amperometry curve under the optimal conditions. The results show that, the detection linear range and detection limit of cyclic voltammetry are 0–10 mM and 1.1 µM and the detection range and detection limit of amperometry method are 10–160 µM and 2.5 µM. The modified electrode shows good selectivity and practicability in water.

Zhang et al. prepared Bacterial cellulose (BC) /GO (BC-GO) composites through embedding BC nanofibers in GO folded sheets by simple ultrasonic mixing method, which was used for nitrite detection [90]. The electrochemical activity of BC-GO modified GCE was the higher than GO, BC and bare GCE and BC/GO = 1:1 to obtain BC-GO composites have the best electrochemical activity. Under the conditions of applied voltage of 1 V, and pH 7, the linear range of the modified electrode for nitrite detection is 0.5–4590 µM, the sensitivity is 527.35 µAµM^−1^cm^−2^ and the detection limit is 0.2 uM. The practicability of modified electrode was verified in tap water and pond water.

The surface of graphene oxide has a large number of oxygen-containing functional groups, such as carboxyl group, hydroxyl group and epoxy group. These functional groups improve the hydrophilicity of graphene oxide and are conducive to chemical modification or functionalization. However, the poor conductivity of graphene oxide limits its direct application in electrochemical sensing. Therefore, it is necessary for graphene oxide to combine with other electroactive and conductive materials for electrochemical detection. The above graphene oxide showed good conductivity and electro catalytic ability in combination with other compounds and showed good performance in the detection of nitrite in water. However, many studies have not verified the repeatability and stability of the composite material, which are the key factors to determine whether the material can be used to make electrochemical sensors.

## 4. RGO-Based Nanocomposites

Reduced graphene oxide (RGO) is the reduction of graphene oxide. The graphene oxide can be reduced by thermal, chemical or electrochemical methods [40]. Electrochemical is the most used method because of simpler and more environmentally friendly [91]. Compared with GO, RGO is hydrophobic but its conductivity and stability are improved. RGO provides a large and stable surface on which nanoparticles and electro catalytic compounds can be fixed repeatedly and beneficial to the electrochemical behavior of adsorption [92]. This section describes RGO combination with other materials for the detection of nitrite and nitrate in water, the performance of each sensors is summarized in Table 3.

### 4.1. Metal

Both metals and RGO have good conductivity and catalytic ability and their combination will be further enhanced these properties. In the detection of nitrate and nitrite in water, Cu, Au and Ag are nano-metal materials which are widely used and they all show excellent detection results.

Cu has low cost and good electro catalytic ability for the reduction of nitrite and nitrate. Zhang et al. studied copper nanoclusters (Cu-NDs) and RGO modified GCE for nitrite reduction detection [93]. First, the glassy carbon electrode was immersed into graphene oxide solution to obtain RGO/GCE by electrodeposition method. Then the RGO/GCE was put into CuSO_4_ solution to obtain Cu-NDs/RGO/GCE by the same method. Under the condition of pH = 2 and detection potential of −0.2V, the linear range of Cu-NDs/RGO/GCE for nitrite detection was 1.25 µM–13 mM, the detection limit was 0.4 µM and the sensitivity was 241 µAmM^−1^cm^−2^. Moreover, the electrode exhibited high selectivity, reproducibility, stability and repeatability.

The mixture of Cu nanoparticles and MWCNT/RGO can simultaneously detect nitrate and nitrite in water [94]. MWCNT/RGO has a large surface and a large number of edges and defects, Cu nanoparticles provide sites of action, which makes the peak current of nitrite and nitrate reduction of modified electrodes significantly increased. Under the condition of pH = 3, the dynamic concentrations of nitrite and nitrate ions were determined by square wave voltammetry (SWV) in the range of 0.1–75 µM with detection limits of 0.03 and 0.02 µM, respectively. The relative standard deviation of this method is less than 2.23% compared with that of Griess method for the determination of mineral water and tap water, which proved that the electrode can be applied to the determination of nitrate and nitrite in practical samples.

Couto et al. prepared Cu/carbon fiber(CF)/RGO thin films by electrodeposition for catalytic nitrate reduction [95]. With the addition of CF and RGO, the three-dimensional electrode has a high specific surface area, which greatly improves the electron transfer ability of Cu particles. Nitrate reduction experiments were carried out in 100 mmL^−1^ K_2_SO_4_/H_2_SO_4_ and 10 mM KNO_3_ solution. They also compared the reduction effects of Cu/RGO/CF electrode at different deposition times (3 s, 6 s and 10 s). The results showed that Cu/RGO/CF electrodes with deposition time of 10 s had good electro catalytic activity for nitrate reduction. This study provides a three-dimensional electrode for monitoring nitrate concentration.

AuNPs have attracted much attention in the field of electroanalytical chemistry due to their unique electro catalytic activity, high surface volume ratio and high stability. Jiao et al. showed the Au-RGO/poly dimethyl diallyl ammonium chloride(PDDA) (Au-RGO/PDDA) nanocomposites was prepared by one-pot method [96]. They have proved that PDDA as reducing and stabilizing agents for this method is fast, simple and cost-effective. At applied voltage of 0.93 V and pH of 6, the Au-RGO/PDDA modified GCE for determined nitrite by differential pulse voltammetry (DPV). The result show linear detection ranges of this modified electrode were 0.005–8.5 µM and 8.5–10,000 µM, the limit of detection is 0.04 µM. The electrode has good stability, reproducibility and selectivity. The average recovery of samples in actual lake water is 100.6%. 

Bimetallic alloys often show better catalysis than individual metal [97]. Li et al. synthesized worm-like Au-Pd/RGO by the simple wet chemical method using polyethylene glycol monooleyl ether as stabilizer, reductant and growth-oriented agent [98]. When the applied potential is 0.85 V and the pH is 7, the nano-material modified glassy carbon electrode is used to detect nitrite. The detection linear range is 0.05–1000 µM and the detection limit is 0.02 µM. The electrode also has fast response, high sensitivity and repeatability. However, NaBr and KI will interfere with the detection results of the electrode and is less stable than other electrodes.

Rajkumar et al. prepared RGO-C60/AuNPs nanomaterials modified glassy carbon electrode for nitrite detection [99]. As Figure 5 show the scheme of preparation of RGO-C60/AuNPs composite. The electro catalytic activity of the modified electrode for nitrite was significantly enhanced from the test result by cyclic voltammetry. Under the optimum working potential of 0.807 V and pH of 5, the linear range of the modified electrode is 0.05–1175.32 µM and the minimum detection limit is 0.013 µM. The electrode also has high selectivity, repeatability and reproducibility. The average recoveries of nitrite in tap, drinking and river water samples were 97.2%, 96.8% and 101.6%, respectively.

The metal-organic framework (MOF) has great catalysis because of its high surface area and open metal sites. He et al. prepared Au/copper-based metal-organic framework(Cu-TDPAT)/ERGO nanomaterials by electrodeposition for nitrite detection [100]. The schematic diagram shows as Figure 6. Compared with Cu-TDPAT/GCE, Cu-TDPAT/ERGO/GCE and bare GCE, Au/Cu-TDPAT/ERGO/GCE has lower impedance, which indicates that Au/Cu-TDPAT/ERGO/GCE has higher conductivity and electron transfer ability. Because of the synergistic effect between the composites, Au/Cu-TDPAT/ERGO/GCE has better electro catalytic effect on NO_2_^—^. Using DPV method, the linear range of the modified electrode for nitrite detection is 0.001–1000 µM and the detection limit is 0.006 µM. The modified electrode has a good detection effect in tap water, which proves that Au/Cu-TDPAT/ERGO composite material is suitable for actual sample analysis.

AuNP-RGO-MWCNTs nanocomposites were synthesized by co-reduction of sodium citrate with ethylene glycol [101]. The synergistic effect leads to the high catalytic activity of the composites for nitrite oxidation and the modified electrode by AuNP-RGO-MWCNTs show good ability to detect nitrite. Under the condition of applied potential of 0.8 V and pH of 5, the detection range is 0.05–2200 µM and the detection limit is 0.014 µM. The electrode has high sensitivity, selectivity and reproducibility and has been successfully applied to the detection of nitrite in local river water.

Silver nanoparticles (AgNPs) have been widely used because of its large specific surface area, good biocompatibility and strong catalytic activity. Ahmad et al. prepared Ag-RGO nanocomposites by microwave-assisted method [102]. The schematic diagram of Ag-RGO material as show in Figure 7. The nanocomposites modified GCE were used to detect nitrite with high sensitivity. Due to the synergistic catalysis of Ag and RGO for nitrite oxidation, the modified electrode showed a high oxidation peak current. The detection linear range of the electrode is 0.1–120 µM, the sensitivity is 18.4 µAµM^−1^cm^−2^ and the detection limit is 0.012 µM. The modified electrode has been proved to be feasible in practical application in pond water.

TiO_2_ has become an attractive electrode material for electrochemical sensor development due to its good biocompatibility, high conductivity and low cost. Zhang et al. synthesized Ag/TiO_2_/RGO nanocomposites for the determination of nitrite concentration [103]. They synthesized TiO_2_/RGO in situ using TiCl_3_ as GO reductant, then NaBH_4_ as reductant obtain Ag/TiO_2_/RGO from AgNO_3_ and Ti/RGO mixed solutions. The electro catalytic ability of nanocomposites modified glassy carbon electrode was studied at pH 7.1 and applied potential 0.8 V. The linear range of detection is 1 µM–1.1 mM, the limit of detection is 0.4 µM and the response time is 2 s. The detection effect in actual tap water and rainwater is also very good.

The combination of AgNPs, poly pyrrole (PPy) and RGO has excellent electro catalytic activity. Kaladevi et al. reported AgNPs/PPy/RGO composites that were prepared by interfacial polymerization using pyrrole organic solvent dispersion as a reducing agent for nitrite detection [104]. It was found that the electrodes modified by the nano-material significantly increased the peak current intensity of nitrite oxidation. The electrode can be used to detect both poisonous hydrazine and nitrite in water. Under the condition of potential 0.7 and pH 7.5, the linear range of nitrite detection is 0.6–6.6 µM and the minimum detection limit is 0.021 µM. The electrode has good applicability in tap water, lake water and drinking water.

Dagci et al. described a novel nitrite detection paper electrode composed of RGO/AgNPs/ Pyronin Y (PyY) hybrid [105]. The schematic diagram of RGO/AgNPs/PyY material as show in Figure 8. At the working voltage of 0.86 V and pH of 5, the electrode showed the detection linear range of 0.1 µM–1 mM with a detection limit of 0.012 µM and a sensitivity of 13.5µAµM^−1^cm^−2^. The electrode also showed excellent selective stability. It was feasible to detect in actual mineral water and lake water. This paper electrode provides a new idea for future detection.

Pt nanoparticles also have excellent electro catalytic activity for nitrite oxidation. Vijayaraj et al. proposed that Pt-electrochemical reduced graphene oxide (ERGO) nanocomposite film modified GCE for nitrite selectivity and sensitivity detection [106]. Platinum nanoparticles were electrochemically and uniformly deposited on the surface of ERGO. It was found that the electro catalytic activity of Pt-ERGO/GCE for nitrite was significantly enhanced. Under the condition of applied potential of 0.75V and pH of 5, the linear range of the modified electrode for nitrite detection is 5–100 µM and 100–1000 µM and the detection limit is 0.22 µM. The detection recovery in tap water is between 95.1 and 103.2%, which proves that the modified electrode can be used in practical samples.

The synergistic effect of metal and RGO improves the electrocatalytic performance of the RGO-based nanocomposites. The electrochemical performance (including detection range, detection limit, sensitivity, etc.) of RGO/metal nanocomposite modified electrode for the detection of nitrate and nitrite was significantly improved.

### 4.2. Metal oxide 

Transition metal oxides (TMOs) are commonly used in electrochemical sensors because of their high surface volume ratio, good conductivity and stability. Deposition on RGO can avoid the dissolution and agglomeration of TMOs to some extent and improve the stability of metal-carbon electrodes [107]. 

Zinc oxide (ZnO) has excellent optical and electrical properties. Marlinda et al. synthesized flower-like ZnO (fZnO) and reduced functionalized graphene oxide (rFGO) nanocomposites by the hydrothermal method [108]. Under the working potential of 0.9 V and pH of 7.2, the ability of fZnO/rFGO modified GCE for nitrite detection was studied. The linear detection range is 10 µM–8 mM and the limit of detection is 33 µM. The electrode has the advantages of fast response, high selectivity, low cost and simple synthesis method. However, the detection limit of this electrode is higher than that of other electrodes.

It seems that Fe_2_O_3_/RGO showed lower LOD for nitrite detection than fZnO/rFGO. Radhakrishnan et al. successfully grafted Fe_2_O_3_ onto RGO nanosheets by one-step hydrothermal method for nitrite detection [109]. The range of electrochemical detection of nitrite through Fe_2_O_3_/RGO material is 0.05–0.78 µM and the detection limit is 0.015 µM. The sensor also shows excellent selectivity and stability. The results of verification in actual tap water show that the sensor can be used in actual environment. 

H-C_3_N_4_ is a two-dimensional material used as a bridge between Fe_2_O_3_ and RGO. Wang et al. described Fe_2_O_3_/H-C_3_N_4_/RGO by hydrothermal method for electrochemical detection of nitrite, in which H-C_3_N_4_ was evenly distributed on the surface of RGO, serving as a bridge connecting Fe_2_O_3_ and RGO and Fe_2_O_3_ as a catalytic center [110]. Because of the synergistic effect of them, the catalytic reaction of Fe_2_O_3_/H-C_3_N_4_/RGO to nitrite was significantly improved. When the potential is 1 V and the pH is 7.4, the detection range of nitrite is 0.025–3000 µM and the detection limit is 0.018 µM. The average recovery of the electrode was 99.27% in actual tap water. However, the applied potential of the modified electrode for nitrite detection is relatively high.

Li et al. synthesized CuO/H-C_3_N_4_/RGO nanocomposites by the hydrothermal method for the detection of nitrite, in which H-C_3_N_4_ is a bridge connecting CuO and RGO [111]. The nanocomposites have larger surface area and faster electron transfer ability, which improves the catalytic ability of nitrite. Under the condition of applied potential of 0.46 V and pH of 8, the linear range and detection limit of CuO/H-C_3_N_4_/RGO modified glassy carbon electrode for nitrite detection were 0.2–100 µM and 0.016 µM, respectively.

With the different experiment conditions, the structure of composite nanomaterials may be different. Yue et al. proposed that CuO_x_/ERGO nanomaterials were synthesized by pH-regulated electrodeposition [112]. The morphology of CuO_x_/ERGO could be controlled by adjusting the pH value. They found that nano-materials with honeycomb morphology had higher electro catalytic activity than those with other shapes, mainly because of their large average electroactive surface area and honeycomb morphology was conducive to electron transfer. When the applied potential is 0.9 V and the pH is 4, the linear range of nitrite detected is 0.1–100 µM and the detection limit is 0.072 µM. The modified electrode also has good selectivity and accuracy in actual tap water.

One-dimensional nanostructured cobalt oxide (Co_3_O_4_) has the advantages of low cost, environmental protection and good conductivity. Co_3_O_4_/RGO nanocomposites show high sensitivity for nitrite detection [113]. Co(NO_3_)_2_.6H_2_O(0.01 mol) and hydrazine benzoate (0.02 mol) were mixed and stirred to prepare precursor complexes. Graphene oxide and precursor complexes were mixed and stirred for 1 h, then reacted at 200 °C for 8 h to obtain Co_3_O_4_/RGO nanocomposites. The Co_3_O_4_/RGO nanocomposites modified GCE for detect nitrite at the applied potential of 0.54 V. The linear range of detection is 1–380 µM, the detection limit is 0.14 µM and the sensitivity is 29.5 µAµM^−1^cm^−2^. The selectivity, reproducibility and stability of the electrode meet the actual needs. The average recovery of the electrode in tap water is 99.8%.

Zhao et al. studied nitrite sensing and detection through Co_3_O_4_-RGO/CNTs modified GCE [114]. Co_3_O_4_ attracts NO_2_^−^ and complexes to produce [Co_3_O_4_ (NO_2_^−^)], which greatly improves the sensitivity. Due to the large surface area and active sites of Co_3_O_4_-RGO/CNTs, the modified electrode has good electro catalysis for the oxidation of nitrite. They optimized parameters such as pH, the amount of modification and the ratio of modified materials. Under the optimum conditions, the linear range of nitrite detection with Co_3_O_4_-RGO/CNTs modified glassy carbon electrode is 0.1 µM–8 mM and the detection limit is 0.016 µM. The detection recovery of the electrode is between 95.7% and 102.2% in actual tap water.

TiO_2_ is a kind of inorganic metal oxide with high isoelectric point and wide band gap. It has been widely used in the preparation of various electrochemical sensors. Li et al. synthesized TiO_2_/RGO nanocomposites with the wet chemical method, which modified GCE for highly sensitive detection of nitrite [115]. TiO_2_ combined with RGO has good electro catalytic activity, which significantly improves the response current to nitrite. When the applied potential is 0.85 V and the pH is 7. TiO_2_/RGO/GCE has the linear range for the detection of nitrite is 1–1000 µM and the detection limit is 0.21 µM. The electrode has been successful application in hot spring water for nitrite detection.

Stankovic et al. compared the electrochemical detection of nitrite with three metal oxides/RGO (TiO_2_/RGO, SeO_2_/RGO, CeO_2_/RGO) [116]. They found the CeO_2_/RGO/GCE had the best catalytic activity for nitrite oxidation. Under the condition of pH 3.4 and working potential 1 V, the linear range of the electrode detection is 07–385 µM and the detection limit is 0.18 µM. Satisfactory results have been obtained from the application of the modified electrode in tap water.

Compared with metal nanoparticles, metal oxide is easy of storage and production and low cost. These studies show the good catalytic activity and chemically reactive facets of metal oxide. This means that RGO/metal oxide modified electrode has low cost and good detection effect in detecting nitrate and nitrite. 

### 4.3. Polymer

Polymers include conductive polymers and polyelectrolytes. Conductive polymers have strong conductivity, which can be enhanced by binding with RGO. Polyelectrolyte has the advantages of low cost, easy film formation and the combination with RGO can attract nitrite ion aggregation. Polymers have been widely used in nitrite electrochemical sensors, such as PPy [104], PDDA [96], PyY [105].

PDDA has strong cations and is attractive to nitrite. Xu et al. prepared PDDA-RGO nanomaterials modified GCE by a simple method for sensitive nitrite detection [117]. PDDA was added to GO solution, then heated to 100 °C and kept for 30 mins to obtain PDDA-RGO nanocomposites. Under the optimum conditions of applied potential 0.75 V and pH 4.5, the detection linear range of the electrode is 0.5 µM–2 mM and the detection limit is 0.2 µM. This electrode has good selectivity and reproducibility. The recovery in actual drinking water is 95.4%–106%. However, PDDA dissolves easily in water solution and the structure of PDDA-RGO will be broken with the use of modified electrode [118]. 

Kesavan et al. prepared reduced graphene oxide modified on GCE by the electrochemical method and then polymerized diaminobenzene (PDAB) onto the electrode surface by electro polymerization to obtain a PDAB/ERGO/glass carbon film modified electrode [119]. Because of the existence of ERGO, the electrode has good conductivity and the positive backbone of PDAB film can attract nitrite ions. Under the condition of applied potential of 0.76 V and pH of 7.2, the linear concentration range of nitrite detected by modified electrode was 7 µM–20 mM and the detection limit is 0.03 µM. The sensor has strong selectivity and high sensitivity, which can be used to detect nitrite in real water samples.

### 4.4. Other Metal Compounds

Other metal complexes also show excellent performance for nitrite detection. 

Transition metal hydroxides have been proved to have high electro catalytic activity [120]. Wang et al. synthesized Ni(OH)_2_/RGO nanocomposites by one-step solvothermal method for selective detection of nitrite [121]. The nano-material has a large surface area and good electro catalytic ability, which improves the catalytic performance of nitrite oxidation. Under the condition of applied voltage 0.9 V and pH 7, Ni(OH)_2_/RGO/GCE has the linear range and detection limit of nitrite were 0.1–663.6 µM and 0.07 µM, respectively. The electrode has fast response, high sensitivity and good reproducibility, the practicability of the composite material has been verified in mineral water and lake water. However, the electrode is not suitable for use in more complex environmental samples.

Transition metal ferricyanide (MHCF) has unique electrochemical properties. Luo et al. prepared cobalt hexacyanoferrate (CoHCF) /RGO (CoHCF-RGO) nanocomposite modified glassy carbon electrodes for the detection of nitrite in water [122]. CoHCF-RGO/GCE was prepared by two times of surface functionalization. The ability of the electrode to quantitatively analyze nitrite was studied by differential pulse voltammetry in phosphate buffer with pH of 6.5. The results show that the linear detection range is 1–100µM and the detection limit is 0.27 µM. The modified electrode also has selectivity, repeatability and reproducibility. Satisfactory results were obtained in actual tap water and river water.

Ferrocene derivatives have redox properties and have good catalytic effect on nitrite oxidation. Rabti et al. fixed chitosan and azidomethyl ferrocene (amFc) covalently on RGO sheets (amFc-RGO/CS), then cast their composite materials on polyester carriers to form screen-printed working electrodes, which can be determined nitrite by cyclic voltammetry and chronoamperometry [123]. At the applied potential is 0.7 V and the pH is 7, the linear concentration of nitrite is 2.5–50 µM and 50–14,950 µM, the sensitivity is 0.00035 µAµM^−1^ and the detection limit is 0.35 µM. The reproducibility, repeatability and long-term stability of the electrode have also been tested in the study.

Metal sulfides are one of the materials used to modify electrodes because of their long-term stability and high conductivity. Balasubramanian et al. studied the preparation of ERGO/beta- cyclodextrin (β-CD)/CdS nanocomposites by one-step electrodeposition, which modified SPCE for the sensitive and selective detection of nitrite [124]. The addition of nano-CdS improves the electro catalytic activity and electron transfer rate of ERGO/β-CD/CdS for nitrite oxidation. In PBS buffer with applied potential of 0.78 V and pH of 7, the linear range of nitrite detection is 0.05–447 µM and the detection limit is 0.021 µM. The electrode has good selectivity, reproducibility and repeatability. The relative standard deviation of detection in tap water and river water is small and it is suitable for practical application.

As a two-dimensional layered structure, MoS_2_ has attracted extensive attention in the field of electrochemistry due to its excellent electro catalytic performance. Hu et al. synthesized RGO-MoS_2_ heterostructure by hydrothermal method and RGO-MoS_2_ modified GCE was used for high sensitivity detection of nitrite [125]. Because of the synergistic effect of RGO and MoS_2_, RGO-MoS_2_/GCE has a fast and efficient electro catalytic ability for nitrite oxidation. When the applied potential is 0.8 V and the pH is 7, the linear detection range of the sensor is 0.2–4800 µM, the detection limit is 0.17 µM and the sensitivity is 0.46µAµM^−1^cm^−2^. The sensor has good selectivity and reproducibility and the detection recovery in actual standard-added tap water is between 98% and 104%.

Polyoxometalates (POMs) have a good catalytic effect on the reduction and oxidation of nitrite, so POMs/RGO-MoS_2_ can improve the detection range of nitrite. Xu et al. prepared 3D phosphomolybdic acid (H_3_PO_4_·12MoO_3_)/MoS_2_/RGO macro porous composites in one pot for nitrite detection [126]. Flower-like spherical MoS_2_ and RGO sheets intertwined to form a three-dimensional macro porous structure. Here, H_3_PO_4_·12MoO_3_ acts as an active material for nitrite oxidation and provides a source of Mo. MoS_2_ enhances the active surface area of the material and RGO enhances the conductivity and catalytic current. H_3_PO_4_·12MoO_3_-MoS_2_-RGO modified GCE has a linear range of 0.5 µM to 8 mM for nitrite detection and the limit of detection is 0.2 µM. The electrode has good selectivity, repeatability, reproducibility and long-term preservation and was proved in local lake water samples.

Three-dimensional macro porous (3D-mp) nanocomposites have more excellent properties. Ma et al. synthesized a novel 3D-mp-RGO-POM by hydrothermal treatment [127]. The 3D-mp-RGO has better conductivity and greater active surface area. Moreover, it greatly promoted mass transfer and effectively reduced the leaching of POM by modified electrodes. Due to the synergistic effect of 3D-mp-RGO and POM, the composite modified GCE has stable and efficient electro catalytic activity for the oxidation of nitrite. When the working potential of the electrode is 0.7 V, the detection linear range is 0.5 µM to 221 µM and 221 µM to 15.221 mM and the detection limit is 0.2 µM. The selectivity, stability and reproducibility of the electrodes were also verified and the results were satisfactory in the actual river water and tap water samples. 

Compared with metal oxide, these metal compounds have special structure, which is conducive to improving the electron transfer and electrochemical activity. Metal compounds combined with the reduced graphene oxide, satisfactory results were obtained in the electro catalysis of nitrate and nitrite.

### 4.5. Others

Other RGO-based materials also improve the performance of detection nitrite and nitrate in water. Mani et al. used hydrazine to obtain chemically reduced graphene oxide (Cr-GO) [128]. This material has more stable properties and is more sensitive to nitrite detection. Under the optimum conditions, the linear concentration range of nitrite detected by Cr-GO/GCE is 8.9–167 µM, the detection limit is 1 µM and the sensitivity is 0.0267 µAµM^−1^. The modified electrode has high repeatability, stability and better antifouling properties and the results of different water samples are satisfactory. However, the preparation of RGO by chemically reduction is more complex and polluted.

Graphene can be used as the ion-to-electron transducer, this property is applied to fabricated a nitrate selective electrode. Tang et al. studied the selectivity and detection ability of Cr-GO modified electrode to nitrate in drinking water [129]. The experimental results show that Cr-GO can significantly promote the work of ion pair transducer and prevent the formation of water layer between ion selective film and graphene layer. The result of this electrode shows the detection range is 10^−4.3^–10^−1^ M, the limit of detection was 3 × 10^−5^ M and the response time of the electrode is less than ten seconds and a reasonable Nernstian slope (57.9 mV per decade of nitrate concentration). This electrode performs well in the detection of nitrate in actual drinking water.

Nitrogen-doped graphene oxide shows good electro catalytic activity for oxygen reduction. Chen et al. developed a non-metallic nitrogen-doped RGO (N-RGO) modified glassy carbon electrode sensor for nitrite detection [130]. Under the condition of oxidation potential of 0.68V and pH of 7, the sensor exhibits the best catalytic activity for nitrite oxidation. The linear range of the sensor is 0.5–5000 µM, the sensitivity is 0.229 µAµM^−1^cm^−2^ and the minimum detection limit is 0.2 µM. The modified electrode has good stability, high selectivity and acceptable reproducibility and the average recovery rate in real river water is 99%.

Multi walled carbon nanotubes (MWCNTs) also have excellent electrical conductivity and are one of the common materials used in electrochemical sensors. Deng et al. synthesized RGO/MWCNTs composite nanomaterials through a simple method for nitrite detection, the structure of nanomaterials show the MWCNTs were encapsulated and covered by RGO [131]. RGO provides many active sites and multi-walled carbon nanotubes provide fast electron transport. When the applied voltage is 0.78 V and the pH is 7, the linear detection range of the RGO/MWCN modified electrode is 0.2–640 µM and the detection limit is 0.07 µM. The recoveries of the modified electrodes were between 93.1% and 105.4% in the standard river water and laboratory wastewater.

The presence of dye molecules in RGO paper structure not only reduces the electro oxidation potential but also improves the effective surface area and has better electro catalytic effect. Aksu et al. prepared flexible, independent RGO paper electrodes doped with phenazine, phenothiazine, phenazine, xanthine, acridine and thiazole class of dyes [132]. By comparing the conductivity and electrochemical performance, acridine (Acr)/RGO paper electrode has the best electro catalytic oxidation performance for nitrite. When the applied potential is 0.985 V and the pH is 5, the linear range of RGO/Acr paper electrode detection is 0.4–3600 µM, the sensitivity is 0.4 µAµM^−1^cm^−2^ and the detection limit is 0.12 µM. The detection results of the electrode in actual mineral water and tap water are similar to those of spectrophotometry. The greatest advantage of this electrode is its flexibility.

Polyhedral oligomeric silsesquioxane (POSS) is a hydrophobic material. POSS/RGO can reduces the oxidation potential and improve hydrophobicity, which is more conducive to the long-term preservation of electrodes. Bai et al. synthesized POSS/RGO nanocomposites by one-step method [118]. The modified GCE with POSS/RGO nanocomposites exhibited good performance in nitrite oxidation catalysis. Under the condition of applied potential of 0.72 V and pH of 7.2, the linear detection range of the modified electrode is 0.5 µM–120mM. The detection limit is 0.08 µM and the sensitivity is 255 µAmM^−1^cm^−2^. The selectivity, stability and reproducibility of the modified electrode have also been verified in this study.

The properties of graphene modified by protoporphyrin IX (PIX)were also improved. Stefan et al. developed two stochastic micro sensor to detect nitrate and nitrite, one electrode is modified by PIX/Gr, another electrode is modified by PIX/RGO [133]. The characteristics of nitrite and nitrate of the same micro-sensor are different, which proves that the micro-sensor can be used to detect nitrate and nitrite in water samples at the same time. These two micro sensors were used in nitrate and nitrite solutions, the micro sensor based on PIX/RGO perform better. The results showed that the linear ranges of nitrite and nitrite were 1.0 × 10^−9^–9.4 × 10^−3^ M/L and the limit of determination is 0.001 µM. It also showed higher sensitivity, selectivity and stability. The results of detecting nitrate and nitrite in different water samples by this sensor are similar to those by ISO standard method. The advantage of this method is that it does not need pretreatment of water sample and can simultaneously detect nitrate and nitrite in water.

Benzyl triethyl ammonium chloride (TEBAC) modified RGO is easier to capture nitrate in water. Chen et al. developed a new nitrate sensor based on RGO/TEBAC [134]. The results showed that the linear detection range of the sensor was 0.0028–28 mg/L, the detection limit was 1.1 µg/L, the response time was within 2–7 s and the selectivity to nitrate ion was high in the solution mixed with other ions. The sensor has high sensitivity and rapidity, have possible applied in real-time monitoring of nitrate. Compared with other field-effect transistor sensors, it has higher stability and lower cost. 

Compared with graphene and graphene oxide, RGO has both conductivity and hydrophilicity with functional groups in its surface and it is easy to combine with other materials. The ability of RGO nanocomposite modified electrode was significantly improved in the application of nitrate and nitrite detection and the modified electrode had good selectivity, repeatability and stability. RGO nanocomposites with 3D structure have larger contact surface and excellent electrical conductivity, which is one of the research directions to improve the catalytic performance of nanomaterials in the future.

## 5. Conclusions and Future Perspectives

This paper review the applications of graphene-based materials in the detection of nitrate and nitrite in water. The advantage and performance of graphene composites, including graphene-based, GO-based and RGO-based in the construction of electrochemical nitrite and nitrate sensors are presented. The detection principles and optimal conditions for the detection of composite modified electrodes are also summarized in relevant tables. 

Graphene-based nanocomposite modified electrode has the high sensitivity and selectivity, fast response rate and excellent stability for nitrate and nitrite detection. Especially, the LOD is reduced to the level of nano-molar and the good selectivity in various water environments. In general, RGO has good conductivity and catalytic activity, which is more useful to synthesize composite materials and detect nitrate and nitrite with electrochemical sensor.

Although the performance of graphene-based composite nano-materials modified electrodes for nitrite detection has been greatly improved, these still exist some challenges for further investigations in electrochemical nitrite and nitrate sensors. Firstly, the performance (LOD and linear range) is still poorer than those of fluorescence method and HPLC. The size, shape, oxidation level, degree of purity, structural defects and degree of dispersion highly affect the electrical conductivity, catalytic activity of graphene-based nanomaterials. Therefore, future research needs to study and synthesize nano materials with 3D structure, large surface area and uniform dispersion to improve their catalytic performance. Secondly, the graphene-based nanocomposites have poor stability and repeatability (after 10 days, the detection current is only 75% of the original), which limits the long-term on-line monitoring of the electrochemical sensor. Therefore, it is necessary to synthesize graphene composite materials with stable connection with other materials and stable fixation on the electrode surface by new methods. Thirdly, these nanomaterials modified electrodes are used in general water or treated water to verify their practicability. For this, the development of real-time electrochemical sensors needs to consider the specific recognition materials for nitrite molecules or the combination of ion selective membrane. The last issue is the high costs of graphene based electrochemical sensors for nitrate and nitrite detection, such as the noble metals/graphene composite. In the future, it needs to stable synthesis the low-cost nanomaterials for large-scale development.

## Figures and Tables

**Figure 1 sensors-20-00054-f001:**
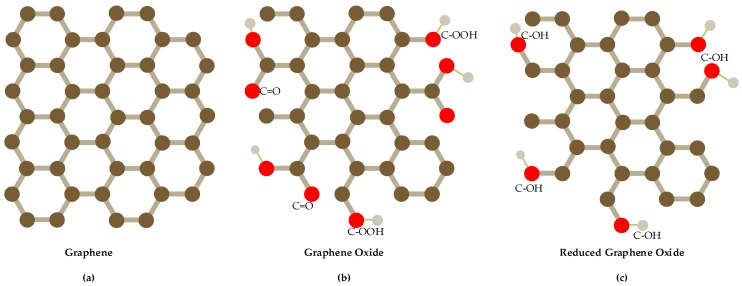
Structures of graphene-based materials show (**a**) the pristine graphene (pure-arranged carbon atoms) with sp^2^-hybridized carbon atoms; (**b**) graphene oxide(GO); (**c**) reduced graphene oxide(RGO).

**Figure 2 sensors-20-00054-f002:**
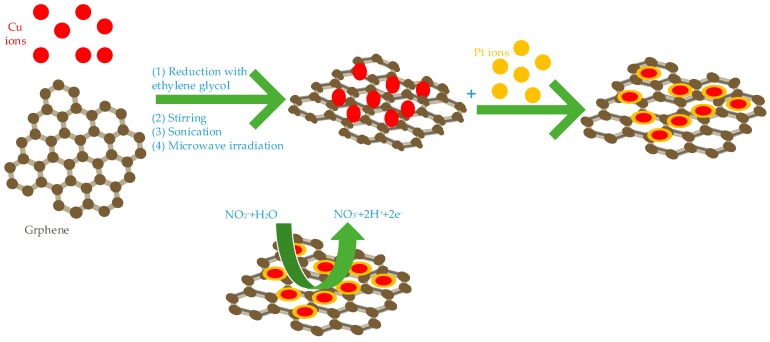
The synthesized progress of Cu@Pt/Gr nanocomposites.

**Figure 3 sensors-20-00054-f003:**
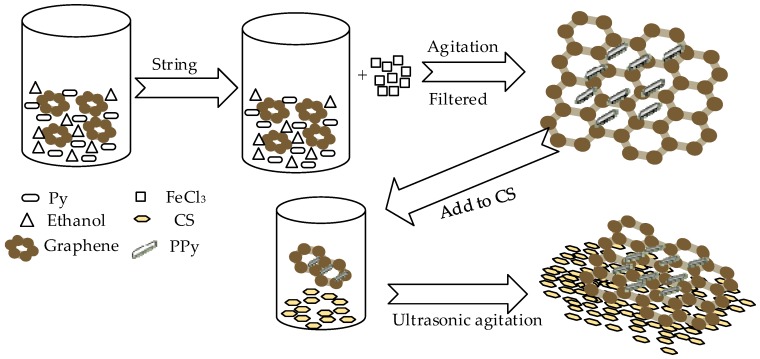
The synthesized progress of Gr/PPy/CS nanocomposites.

**Figure 4 sensors-20-00054-f004:**
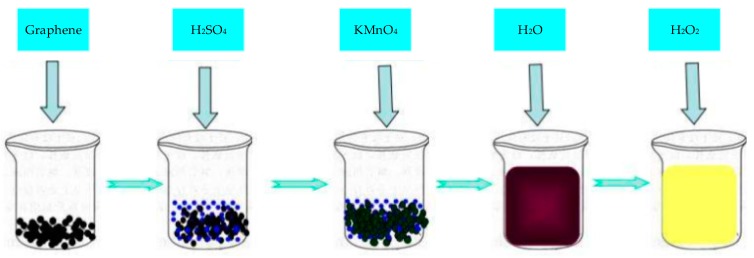
Process for synthesis of graphene oxide (GO) by hummers method.

**Figure 5 sensors-20-00054-f005:**
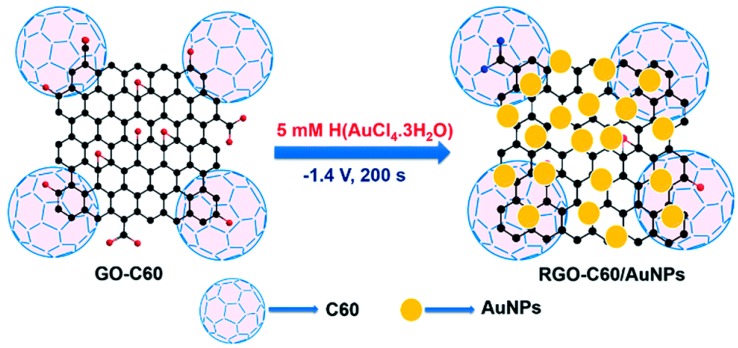
Scheme of preparation of RGO-C60/AuNPs (Reprinted from Reference 96 with permission from ROYAL SOCIETY OF CHEMISTRY, copyright 2016).

**Figure 6 sensors-20-00054-f006:**
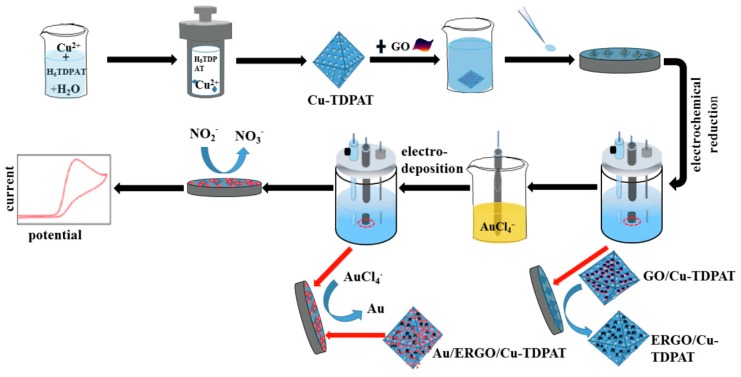
Schematic diagram of Au/ERGO/Cu-TDPAT for nitrite detection(Reprinted from Reference [97] with permission from ELSEVIER, copyright 2019).

**Figure 7 sensors-20-00054-f007:**
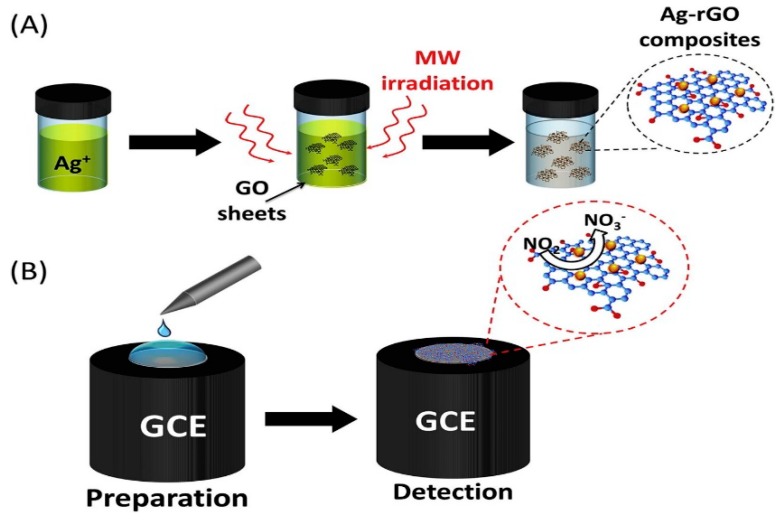
Schematic diagram of Ag/RGO for nitrite detection(Reprinted from Reference [99] with permission from ELSEVIER, copyright 2018).

**Figure 8 sensors-20-00054-f008:**
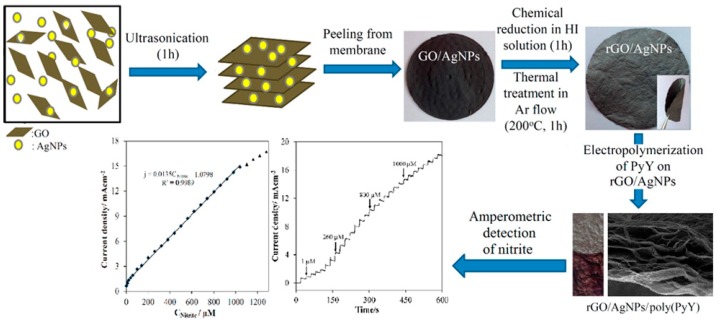
Schematic diagram of RGO/AgNPs /PyY for nitrite detection(Reprinted from Reference [102] with permission from American Chemical society, copyright 2016).

**Table 1 sensors-20-00054-t001:** Graphene-based material for nitrate and nitrite detection.

Material	Method	Detection Conditions	Detection Range(µM)	LOD (µM)	Sensitivity (µAµM^−1^cm^−2^)	Long-term Stability	Repeatability (RSD)	Reproducibility	Ref
Gr-MWCNTs/FeNPs	Amperometry	0.77V pH = 5	0.1–1680	0.076	0.697	94.3%/3 W	3.55%	2.25%	[43]
NPCF-GNs/GCE	Amperometry	0.8 V pH = 9	0.1–100	0.088	3.1	100%/1 h	3.74%	5.14%	[44]
Gr/CuNPs	DPV	−0.8V	10–90	7.89	N/A	N/A	N/A	N/A	[45]
CoNPs-PEDOT-Gr/GCE	Amperometry	0.45 V pH = 4.5	0.5–240	0.15	N/A	90%/30 D	3.1%	4.4%	[46]
Cu@Pt/Gr	Chronoamperometry	0.85 V pH = 4	1–1000 1000–15000	0.035	0.021 0.063	86.9%/1 M	3.2%	4.1%	[47]
Ni@Pt/Gr	Amperometry	0.85 V pH = 4	283–1230 10–15000	0.49	0.192 0.085	90.2%/3 W	3.3%	3.4%	[48]
Co@Pt/Gr	Amperometry	0.85 V pH = 4	1–2000 2000–15000	0.145	0.046 0.098	87.1%/1 M	4.2%	3.9%	[49]
AuNPs/SG/GCE	Amperometry	0.73 V	10–3960	0.2	0.454^a^	93.5%/1 M	1.92%	2.86%	[50]
Au/Gr/MCE	DPV	0.74 V pH = 4.75	0.3–720	0.1	N/A	N/A	N/A	N/A	[51]
w-Gr/Au/GCE	DPV	0.84 V pH = 7.2	0.1–1000 1000–5000	0.06	N/A	95%/7 D	N/A	4%	[52]
AuNP/MnO_2_-Gr-CNT/GCE	Amperometry	0.8 V pH = 7	1–2896	0.05	0.181	94%/1 M	N/A	4.6%	[53]
Gr/PPy/CS/GCE	Amperometry	0.9 V pH = 4	0.5–722	0.1	N/A	85%/1 M	1.8%	N/A	[54]
CG/PPy/CS/GCE	DPV	0.876 pH = 4	0.2–1000	0.02	N/A	97.1%/18 D	3.4%	1.2%	[55]
PEDOT/Gr	Amperometry	0.78 pH = 6	0.3–600	0.1	N/A	95%/35 D	3.1%	3.9%	[56]
PEDOT-Gr/Ta	DPV	0.7 V pH = 7	20–2000	7	N/A	100%/2 M	4.5%	7.8%	[57]
Fe_1.833_(OH)_0.5_O_2.5_/ PDDA /NG/GCE	Amperometry	0.837V pH = 7.4	0.1–347 347–1275	0.027	N/A	99.5%/30 D	0.65%	0.87%	[58]
PW_12_O_40_^3−^/CS-Gr/CY-AuE	Amperometry	−0.18 V	0.18–6740	0.02	N/A	92.2%/2 M	3.26%	4.56%	[60]
MOF-525/Gr	Amperometry	0.85 V	100–2500	0.75	0.0938	N/A	N/A	N/A	[61]
NG-CuS/GCE	Amperometry	0.89 V pH = 6	0.1–14020	0.033	N/A	94.3%/15 D	3.2%	6%	[62]
Gr/CuE	Amperometry	−1.28 V pH = 12	9–940	10	N/A	N/A	N/A	N/A	[63]
GAs /NO_3_^−^-ISE/GCE	CV	N/A	10–100000	10^1.8^	N/A	NS/21 D	N/A	N/A	[65]
Gr/CD/SPCE	Amperometry	0.81 pH = 5	0.7–2250	0.26	0.476	N/A	3.2%	2.1%	[66]
Gr/GCE	Amperometry	0.805 V pH = 5	1–250	0.24	0.843	80%/15 D	N/A	N/A	[67]
GNs/GCE	Amperometry	0.9 V pH = 3	05–45 45–105	0.22	6.3219	97.6%/5 D	1.9%	3.3%	[68]
ERHG/GCE	Amperometry	0.92 V pH = 7.4	0.2–10000	0.054	0.311	95%/21 D	2.23%	2.34%	[69]
3D-HG	DPV	0.816 pH = 4	0.05–500 500–10000	0.01	0.764 0.353	96.5%/3 W	1.85%	2.1%	[70]
NGQDs@NCNFs	DPV	0.82 V pH = 4.5	5–300 400–3000	3	N/A	N/A	3.02%	4.27%	[71]

Notes: ^1^ DPV; Differential Pulse Voltammetry, ^2^ LSV; Linear Sweep Voltammetry, ^3 a^: uAuM^−1^, ^4^ S: second, ^5^ D: Day, ^6^ C: cycles, ^7^ M: Month, ^8^ W; Week, ^9^ NSD; No Significant Difference, ^10^ N/A: Not Applicable, ^11^ CV: Cyclic voltammetry.

**Table 2 sensors-20-00054-t002:** GO-based material for nitrate and nitrite detection.

Material.	Method	Detection Conditions	Detection Range(µM)	LOD (µM)	Sensitivity (µAµM^−1^cm^−2^)	Long-term Stability	Repeatability (RSD)	Reproducibility (RSD)	Ref
Cube-like Pt/GO	Amperometry	1.02 V pH = 4	0.5–227780	0.2	0.358^a^	N/A	N/A	N/A	[73]
Aggregation-like Pt/GO	Amperometry	1 V pH = 4	0.85–227740	0.02	0.523^a^	N/A	N/A	N/A	[73]
Globular-like Pt/GO	Amperometry	1.05 pH = 4	0.05–177780	0.015	0.405^a^	N/A	N/A	N/A	[73]
GO-Ag/GCE	LSV, Amperometry	0.94 V pH = 7	10–180 1–1000	2.1 0.037	N/A	N/A	N/A	N/A	[74]
AgNPs/GO/GCE	LSV	0.91 V pH = 7.2	1–1000	0.24	N/A	N/A	N/A	N/A	[75]
Ag-Fe_3_O_4_-GO	Amperometry	0.8 V 6	0.5–720 720–8150	0.17	1.966 0.426	94%/1 M	N/A	3.68%	[76]
Ag-AEfG	Amperometry	0.85 V pH = 7.4	0.05–3000	0.023	0.2	98.7%/300 C	N/A	2.03%	[77]
GO-MWCNTs-PMA-Au	DPV	0.73 V pH = 6	2–10000	0.67	0.484	91.2%/4 W	N/A	4.0%	[79]
Au/GO-CS/GCE	Amperometry	0.8 V pH = 5	0.9–18.9	0.3	N/A	N/A	N/A	N/A	[80]
Au-HNTs-GO	Amperometry	0.8 V pH = 6	0.1–6330 6330–61900	0.03	0.0231 0.0865	93%/1 W	3.5%	N/A	[81]
PANI/GO	Amperometry	0.83 V pH = 5	2–44000	0.5	0.117	N/A	2.4%	2.2%	[82]
Mn_3_O_4_ MC/GO/SPE	Amperometry	0.7 V pH = 7	0.1–420 420–1318	0.02	2.37 1.23	97.8%/30 D	N/A	4.82%	[83]
MnO_2_/GO-SPE	DPV	0.55 V pH = 7.4	0.1–1 1–1000	0.09	1.25 0.005	N/A	N/A	N/A	[84]
Pt-GO-PB	LSV	pH = 7	N/A	6.6	0.0084^a^	82.8%/20 C	N/A	N/A	[85]
Pt-GO-Fe_2_O_3_	LSV	pH = 7	N/A	16.58	0.0093^a^	72.5%/20C	N/A	N/A	[85]
Fe_3_O_4_/GO/COOH/GCE	DPV	pH = 4	1–85 90–600	0.37	0.192^a^	N/A	N/A	N/A	[86]
CS/FEPA-GO/GCE	Amperometry	0.756 V	0.3–3100	0.1	N/A	92%/1M	3.42%	N/A	[87]
Gem-GO-POM/GCE	DPV	0.6 V	5–500	0.39	0.021^a^	N/A	N/A	NSD	[88]
GO/MnNH_2_TPP/GCE	Amperometry	0.76 pH = 4	10–160	2.5	N/A	N/A	N/A	N/A	[89]
BC-GO	Amperometry	1 V pH = 7	0.5–4590	0.2	527.35	91%/1M	2.75%	N/A	[90]

Notes: ^1^ DPV; Differential Pulse Voltammetry, ^2^ LSV; Linear Sweep Voltammetry, ^3 a^: µAµM^−1^, ^4^ S: second, ^5^ D: Day, ^6^ C: cycles, ^7^ M: Month, ^8^ W; Week, ^9^ NSD; No Significant Difference,^10^ N/A: Not Applicable

**Table 3 sensors-20-00054-t003:** RGO-based material for nitrate and nitrite detection.

Material	Method	Detection Conditions	Detection Range(µM)	LOD (µM)	Sensitivity (µAµM^−1^cm^−2^)	Long-term Stability	Repeatability	Reproducibility	Ref
CuNDs/RGO	Amperometry	−0.2 V pH = 2	1.25–13000	0.4	0.214	87%/4 W	4.2%	3.3%	[93]
Cu/MWCNT/RGO	SWV	pH = 3	0.1–75	0.03	N/A	N/A	N/A	N/A	[94]
Au-RGO /PDDA/GCE	DPV	0.93 V pH = 6	0.05–8.5	0.04	N/A	97.8%/2 W	1.9%	N/A	[96]
Au-Pd/RGO/	Amperometry	0.85 V pH = 7	0.05–1000	0.02	N/A	75%/10 D	2.67%	N/A	[98]
RGO/C60/AuNPs/GCE	Amperometry	0.807 V pH = 5	0.05–1175.32	0.013	N/A	96.2%/2 D	4.1%	5.7%	[99]
Au/Cu-TDPAT/ERGO/GCE	DPV	0.77 V pH = 7	0.001–1000	0.006	N/A	95%/6 D	N/A	5%	[100]
AuNP/RGO/MCNT/GCE	Amperometry	0.8 V pH = 5	0.05–2200	0.014	1.201	NSD/30 D	N/A	4.1%	[101]
Ag-RGO/GCE	DPV	pH = 7.4	0.1–120	0.012	18.4	94.5%/5 W	N/A	2.38%	[102]
Ag/TiO_2_/rGO/GCE	Amperometry	0.8 V pH = 7.1	1–1100	0.4	0.112	90%/3 W	N/A	5%	[103]
AgNPs@PPy/rGO	DPV	0.7 V pH = 7.5	0.6–6.6	0.021	N/A	98.3%/250 C	N/A	2.2%	[104]
rGO/AgNPs/poly (PyY)/	Amperometry	0.86 V pH = 5	0.1–1000	0.012	13.5	65%/3 M	N/A	1.1%	[105]
Pt-ErGO/GCE	Amperometry	0.75 V pH = 5	5–100 100–1000	0.22	N/A	98.98%/12 D	5.94%	1.02%	[106]
fZnO/rFGO/GCE	Amperometry	0.9 V pH = 7.2	10–8000	33	0.38^a^	N/A	N/A	N/A	[108]
Fe_2_O_3_-rGO/GCE	DPV	pH = 7	0.05–780	0.015	0.204	98.1%/10 D	2.2%	1.23%	[109]
Fe_2_O_3_/H-C_3_N_4_/RGO	Amperometry	1 V pH = 7.4	0.025–3000	0.0186	0.0487^a^	98%/15 D	0.74%	1.65%	[110]
CuO/H-C_3_N_4_/RGO	CV	0.46 V pH = 8	0.2–110	0.016	0093^a^	97.6/100 C	N/A	N/A	[111]
CuO_x_/ERGO	Amperometry	0.9 V pH = 4	0.1–100	0.072	N/A	N/A	N/A	N/A	[112]
Co_3_O_4_/RGO	Amperometry	0.54 V	1–380	0.14	29.5	N/A	N/A	1.93%	[113]
Co_3_O_4_-RGO/CNTs	Amperometry	0.8 V pH = 7	0.1–8000	0.016	0.408	83.3%/1M	N/A	N/A	[114]
TiO_2_/RGO	DPV	0.85 V pH = 7	1–1000	0.21	N/A	N/A	N/A	N/A	[115]
CeO_2_/RGO/GCE	Amperometry	1 V pH = 3.4	0.7–385	0.18	N/A	N/A	3.8%	4.2%	[116]
PDDA-RGO	Amperometry	0.75 V 4.5	0.5–2000	0.2	N/A	N/A	3.9%	7.3%	[117]
PDAB/ERGO/GCE	Amperometry	0.76 V pH = 7.2	7–20000	0.03	N/A	94.8%/30 D	5.2%	4.3%	[119]
Ni(OH)_2_/RGO	Amperometry	0.9 V pH = 7	0.1–663.6	0.07	21.93^a^	90%/10 D	2.4%	4.2%	[121]
CoHCF/RGO/GCE	DPV	6.5	1–100	0.27	N/A	N/A	5.5%	7.3%	[122]
amFc-RGO/CS/	Amperometry	0.7 V pH = 7	2.5–50 50–14950	0.35	0.00035^a^	87%/4 W	6%	7.7%	[123]
ERGO/β-CD/CdS/SPCE	Amperometry	0.78 V pH = 7	0.05–447	0.021	0.00336	95.1%/25 D	2.99%	2.13%	[124]
RGO-MoS_2_/GCE	Amperometry	0.8 V pH = 7	0.2–4800	0.17	0.46	N/A	N/A	1.2%	[125]
H_3_PO_4_·12MoO_3_/MoS_2_/ RGO/GCE	Amperometry	0.87 V	0.5–8000	0.2	0.379	90%/1 M	3.5%	5.1%	[126]
3D-mp-RGO/POM/GCE	Amperometry	0.7 V	0.5–221 221–15221	0.2	N/A	94%/1 M	2.5%	4.7%	[127]
CRGO/GCE	Amperometry	0.8 V pH = 5	8.9–167	1	0.0267^a^	92.7%/2500 S	0.726%	2.29%	[128]
CRGO/NO_3_^−^/GCE	CV	N/A	10^−1.3^–10^2^	30	N/A	N/A	N/A	N/A	[129]
N-RGO/GCE	Amperometry	0.68 V pH = 7	0.5–5000	0.2	0.229	93.8%/1 M	1.5%	6.2%	[130]
RGO/MWCN/GCE	Amperometry	0.78 V pH = 7	0.2–640	0.07	N/A	96%/4 W	N/A	5.8%	[131]
RGO/Acr	Amperometry	0.985 V pH = 5	0.4–3600	0.12	0.4	84%/2 M	N/A	1.1%	[132]
POSS/RGO	Amperometry	0.72 V pH = 7.2	0.5–120000	0.08	0.255	N/A	N/A	5%	[118]
PIX/RGO	N/A	N/A	0.001–1000	0.001	19900 s^−1^M	98.2%/10 D	N/A	N/A	[133]
RGO/TEBAC	N/A	N/A	0.0028–28 mg/L	1.1 ug/L	N/A	N/A	N/A	N/A	[134]

*Notes*: ^1^ DPV; Differential Pulse Voltammetry, ^2^ LSV; Linear Sweep Voltammetry, ^3 a^: µAµM^−1^, ^4^ S: second, ^5^ D: Day, ^6^ C: cycles, ^7^ M: Month, ^8^ W; Week, ^9^ NSD; No Significant Difference, ^10^ N/A: Not Applicable

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
