# Peer review of "Application of Graphene-Based Materials for Detection of Nitrate and Nitrite in Water—A Review"

_sensors, 2019, doi:10.3390/s20010054_

Round 1

Reviewer 1 Report

The manuscript is good structured and adequately conceived. However, a major revision is required.  Therefore, some points need to be considered, which are the following:
Abstract:

The abstract is the main attraction for a paper. In this manuscript, the abstract is very weak and it contains a lot of errors. It needs new reformulation.

Introduction:

-Some parts in the introduction needs to have reference, example line 42-43.

- The reference placement is wrong; it needs to be before any punctuation marks.

- The used abbreviation should be identified, then it is possible to use.

Body of the manuscript:

-The text contains errors concerning the limit of detection range. Therefore, all the ranges should be checked (eg. see line 140).

-The units need to be repeated in the whole text (eg. µM not uM).

-The first use of each abbreviation comes with the full terminology.

-In whole manuscript, there are a lot of typos mistakes.

Conclusion:

The conclusion should be reformulated to be more compact with illustrating more about the main advantageous behind the use of graphene-based materials in the detection of nitrate and nitrite in water. Limitations of these sensors should be also mentioned with providing some solutions that can be considered as future work.

Author Response

Thanks for your comments. We have revised this paper point to point according to your comments. The follows are our reply to your comments

Point 1: The abstract is the main attraction for a paper. In this manuscript, the abstract is very weak and it contains a lot of errors. It needs new reformulation.

Response 1: we have addressed this point by re-writing the abstract to significantly improve it.

Introduction:

Point 2: Some parts in the introduction needs to have reference, example line 42-43.

Response 2: Following the reviewer’s feedback, we have added relevant references in the introduction, such as line43-44, line 54-55, line 56-57, line 65-67, line 71-73.

Point 3: The reference placement is wrong; it needs to be before any punctuation marks.

Response 2: We apologize for these errors. The reference placement and format have been revised to address this requirement.

Point 4: The used abbreviation should be identified, then it is possible to use.

Response 2: We have reviewed and revised the entire article to define all abbreviations for the first used.

Body of the manuscript:

Point 5: The text contains errors concerning the limit of detection range. Therefore, all the ranges should be checked (eg. see line 140).

Response 2: We apologize for the mistakes. We have checked all the parameters in the table and corrected the mistakes.

Point 6: The units need to be repeated in the whole text (eg. µM not uM).

Response 2: We have checked and corrected the writing of all these units.

Point 7: The first use of each abbreviation comes with the full terminology.

Response 2: We have reviewed and revised the entire article to define all abbreviations for the first used.

Point 8: In whole manuscript, there are a lot of typos mistakes.

Response 2: We have carried out a thorough proofreading to eliminate typos and errors.

Conclusion:

Point 9: The conclusion should be reformulated to be more compact with illustrating more about the main advantageous behind the use of graphene-based materials in the detection of nitrate and nitrite in water. Limitations of these sensors should be also mentioned with providing some solutions that can be considered as future work.

Response 2: The conclusion has been significantly improved. Firstly, we have summarized the content of this paper, then introduced the advantages of graphene-based electrochemical sensors, and finally discussed the shortcomings and provided recommendations for future research.

Please see the revised section as highlighted in the main manuscript.

Reviewer 2 Report

In general, this review systematically summarizes the application of graphene-based materials for the determination of nitrate and nitrite in water. However, there are some minor errors that need to be noticed and corrected. In addition, the summary and discussion are needed in every section.

[Lack of summary and discussion]

The author simply gives examples and describes related works, and lacks the corresponding summary and comments in every section.

[minor errors]

In line 118, the unit of nitrate concentration is confusing, and there are many similar errors in this manuscript, please check and correct them carefully. The format of the references needs to be unified Some related works (such as Trends in Analytical Chemistry, 2018, 105: 75; 2019, 113, 1. Biosensors & Bioelectronics, 2015, 64: 373. Food Chemistry 2019, 274: 8) should be added in this review. The writing needs to be modified.

Author Response

Thanks for your comments. We have revised this paper point to point according to your comments. The follows are our reply to your comments

Reviewer 2

Point 1Lack of summary and discussion: The author simply gives examples and describes related works, and lacks the corresponding summary and comments in every section.

Response 1: We have addressed this comment by adding a summary at the end of each subsection and main section.

Detailed changes are highlighted in the main manuscript.

Point 2: In line 118, the unit of nitrate concentration is confusing, and there are many similar errors in this manuscript, please check and correct them carefully.

Response 2: we have modified all the unit symbols in the revised paper and presented them with the correct formats.

Point 3: The format of the references needs to be unified

Response 3: we have checked and revised the format of references in the article

Point 4: Some related works (such as Trends in Analytical Chemistry, 2018, 105: 75; 2019, 113, 1. Biosensors & Bioelectronics, 2015, 64: 373. Food Chemistry 2019, 274: 8) should be added in this review.

Response 4: we have read the suggested references and added them to the references, there are ref.43, ref.27, ref.40, ref.25 respectively.

Point 5: The writing needs to be modified.

Response 5: The writing has been improved with the help of an external professor.

Round 2

Reviewer 1 Report

- In the part of RGO,the first sentence needs to be changed:

"Reduce graphene oxide (RGO) is a necessary deoxidization treatment to GO"-> The reduction of graphene oxide...

-The term "reduced graphene oxide" should be verified. It is not reduce graphene oxide.

Example:"Pt-electrochemical reduce graphene oxide"-"Pt-electrochemical reduced graphene oxide"

Response: We have corrected the mistakes according to the reviewer's comments. The sentence of 'Reduce graphene oxide (RGO) is a necessary deoxidization treatment to GO" was revised to “Reduced graphene oxide (RGO) is the reduction of graphene oxide", We changed all reduce graphene oxide to reduced graphene oxide.